Improvement strategies for heuristic algorithms based on machine learning and information concepts: a review of the seahorse optimization algorithm

Zheng Shixing b22140221@njupt.edu.cn
School of Economics, Nanjing University of Posts and Telecommunications , Nanjing, Jiangsu , China
Alatas Bilal
Electronic publication date: 2025 Apr 1
Publication date: 2025
Volume: 11
Electronic Location ID: e2805
Received 2024 Nov 28; Accepted 2025 Mar 13
Copyright: © 2025 Zheng
Copyright year: 2025
Copyright holder: Zheng
License: This is an open access article distributed under the terms of the Creative Commons Attribution License, which permits unrestricted use, distribution, reproduction and adaptation in any medium and for any purpose provided that it is properly attributed. For attribution, the original author(s), title, publication source (PeerJ Computer Science) and either DOI or URL of the article must be cited.
License URL: https://creativecommons.org/licenses/by/4.0/

Keywords: Entropy crossover strategy, Metaheuristic algorithm, Seahorse optimization, Swarm intelligence, Logistic-KNN inertia weights, Statistic, Machine learning

Funding: The authors received no funding for this work.

==============================
To overcome the mechanical limitations of traditional inertia weight optimization methods, this study draws inspiration from machine learning models and proposes an inertia weight optimization strategy based on the K-nearest neighbors (KNN) principle with dynamic adjustment properties. Unlike conventional approaches that determine inertia weight solely based on the number of iterations, the proposed strategy allows inertia weight to more accurately reflect the relative distance between individuals and the target value. Consequently, it transforms the discrete “iteration-weight” mapping ( t→w) into a continuous “distance-weight” mapping ( d→w), thereby enhancing the adaptability and optimization capability of the algorithm. Furthermore, inspired by the entropy weight method, this study introduces an entropy-based weight allocation mechanism in the crossover and mutation process to improve the efficiency of high-quality information inheritance. To validate its effectiveness, the proposed strategy is incorporated into the Seahorse Optimization Algorithm (SHO) and systematically evaluated using 31 benchmark functions from CEC2005 and CEC2021 test suites. Experimental results demonstrate that the improved SHO algorithm, integrating the logistic-KNN inertia weight optimization strategy and the entropy-based crossover-mutation mechanism, exhibits significant advantages in terms of convergence speed, solution accuracy, and algorithm stability. To further investigate the performance of the proposed improvements, this study conducts ablation experiments to analyze each modification separately. The results confirm that each individual strategy significantly enhances the overall performance of the SHO algorithm.

Introduction

Non-convex function optimization presents a complex challenge due to the presence of numerous local optima, often resulting in convergence to suboptimal solutions rather than the global optimum. These optimization problems are of critical importance across various fields. In data mining, for instance, model training fundamentally involves minimizing a complex, non-convex loss function (So et al., 2020). In financial engineering, non-convex optimization plays a role in portfolio optimization and option pricing (Lou, 2023). Similarly, in industrial design, the optimization of components or product designs is often formulated as a non-convex optimization problem (Yu et al., 2025). Additionally, non-convex optimization is used in signal processing to enhance the performance of image processing and communication systems (Vlaski & Sayed, 2021).

To address these challenges, numerous heuristic algorithms have been proposed in recent years. Inspired by natural phenomena and biological behaviors, these algorithms aim to solve complex optimization problems and have shown significant performance in large-scale searches and complex structural optimization. Heuristic algorithms can generally be categorized into four types: swarm intelligence-based algorithms, evolutionary algorithms, physics-inspired algorithms, and human behavior-based algorithms. Representative algorithms within each category are listed in Table 1. These methods have achieved remarkable success across various domains as powerful tools for non-convex optimization. However, issues such as susceptibility to local optima, slow convergence speed, lack of population diversity, and premature convergence remain prevalent.

Table 1 Classification of meta-heuristic algorithms with inspiration sources.

Classification	Algorithm name	Inspired by	
Swarm intelligence-based	PSO (Kennedy & Eberhart, 1995)	Flocking behavior of birds and fish	
	ACO (Dorigo, Birattari & Stutzle, 2006)	Ants’ pheromone-based path optimization	
	ABC (Okwu & Tartibu, 2021)	Bees’ foraging strategy	
	GWO (Mirjalili, Mirjalili & Lewis, 2014)	Wolves’ cooperative hunting	
	DMOA (Agushaka, Ezugwu & Abualigah, 2022)	Collective hunting in dwarf mongooses	
	AO (Abualigah et al., 2021b)	Eagles’ hunting and vision	
	SHO (Zhao et al., 2023)	Seahorses’ adaptive habitat behavior	
Evolutionary algorithms	GA (Holland, 2012)	Natural selection and evolution	
	DE (Storn & Price, 1997)	Differential mutation for diversity	
Physics-inspired	SA (Rutenbar, 1989)	Simulated annealing process	
	GSA (Kalivas, 1992)	Gravitational interactions	
Human and mathematics-based	TS (Khalid et al., 2024)	Tabu list from human memory	
	AOA (Abualigah et al., 2021a)	Arithmetic operations in optimization	

To address these limitations, various improvement strategies have been proposed by researchers. For example, the concept of inertia weight has been widely adopted in particle swarm optimization (PSO) to regulate particle velocity updates within the search space. Gu et al. (2022) introduced the inverted S-shaped inertia weight (ASCSO-S) algorithm, which utilizes an S-shaped function to adaptively adjust the inertia weight across different stages of the algorithm. Specifically, the S-shaped inertia weight starts with a relatively high value during the initial iterations and gradually decreases as the iterations progress, enabling the algorithm to dynamically adjust its “exploration speed” throughout the optimization process. Compared to linearly decreasing inertia weight, the non-linear decrement characteristic of the S-shaped weight allows for flexible adjustment of the decrement rate based on different optimization scenarios, thereby enhancing the algorithm’s generalization capability. Furthermore, in the hybrid particle swarm optimization based on intuitionistic fuzzy entropy (IFEHPSO) algorithm, Wang et al. (2021) utilized the concept of intuitionistic fuzzy entropy from fuzzy mathematics to introduce adaptive perturbations into the inertia weight. This approach departs from the conventional “function-based” weight framework and focuses on the aggregation degree of the population. By using intuitionistic fuzzy entropy to measure population aggregation (with lower aggregation leading to higher entropy and higher aggregation resulting in lower entropy), this method aligns with the PSO algorithm’s iterative dynamics, where exploration is more intense in the early stages and diminishes over time. As a result, intuitionistic fuzzy entropy allows for more intelligent, real-time adjustments of the inertia weight during iterations, further enhancing the algorithm’s performance in solving complex optimization problems.

In addition, improving initialization strategies to enhance population diversity has been proven to be a crucial research focus in optimization algorithms. In the Cauchy Gray Wolf Optimizer (CGWO) algorithm, Li & Sun (2023) proposed an initialization strategy based on the Cauchy distribution, while the authors of the Tent-enhanced Aquila Optimizer (TEAO) introduced an initialization strategy based on the Tent map (Fu et al., 2024). By refining the initialization phase, these methods ensure a more random distribution of the population, providing broader global information coverage during the early stages of the algorithm and strengthening its global search capability.

At the same time, update strategies for the exploration phase have also been extensively studied. For instance, Hu et al. (2022b) incorporated fast-moving and crisscross mechanisms into the enhanced sine cosine algorithm (QCSCA) algorithm, significantly improving its performance during the exploration phase and enhancing its precision in global searches. Additionally, Hu et al. (2022a) integrated multiple optimization algorithms within the improved golden eagle optimizer (IGEO) framework, leveraging the strengths of various strategies to achieve a better balance between exploration accuracy and convergence speed, thereby further improving the overall performance.

These improvement strategies have significantly enhanced the algorithm and provided new insights into the innovation of metaheuristic algorithms. In particular, they offer valuable perspectives on how to effectively adjust individual search strategies and optimize the update of inertia weights, which serve as the motivation for further research in this article.

In swarm intelligence optimization algorithms, the degree of population aggregation directly affects the search efficiency. When the distance between individuals is too small, it may lead to premature convergence, restricting the search space and affecting global optimization capability. On the other hand, if individuals are too widely dispersed, search efficiency may decrease, and computation time may increase. Therefore, properly controlling the aggregation of the population is crucial to improving algorithm performance. Similarly, the K-nearest neighbor (KNN) algorithm is closely related to the distance between individuals. KNN is a non-parametric classification method that determines the nearest neighbors of a target data point by calculating the distances between the target and the sample data, and then classifies the target based on the category labels of its neighbors. Moreover, the inference process of KNN relies entirely on sample data, effectively reflecting the local distribution characteristics of the data.

Similarly, logistic regression, as a classic S-shaped curve function, has a simple structure that is easy to interpret, and similar ideas can be applied in the optimization process. In logistic regression, introducing the concept of S-shaped inertia weights can enhance the optimization process. Inertia weights adaptively adjust the update step size during the algorithm’s iterations, balancing the exploration of global optima and the exploitation of local optima, thus further improving the algorithm’s search performance.

At the same time, the principle of information entropy provides another potential optimization approach. Information entropy is often used to quantify the informational disparity between indicators and determine their weights accordingly. The core idea is that a lower entropy value means the indicator contains more significant information, exhibits greater variability, and has a more substantial impact on the comprehensive evaluation, thus deserving a higher weight. Conversely, a higher entropy value indicates the indicator provides less information and should be assigned a lower weight. This method, similar to the update of inertia weights, can further optimize the weight allocation process, thereby enhancing the algorithm’s exploration and search efficiency.

In recent years, many researchers have combined machine learning with intelligent optimization algorithms. For example, Ran et al. (2024) integrated genetic algorithms (GA), Ant Colony Optimization (ACO), and K-means clustering to enhance performance; Long et al. (2025) employed ACO to optimize reinforcement learning models, providing a novel approach for solving QCSP problems; Hemdan et al. (2025) combined GA with ensemble learning, offering new insights for real-time fault detection.

However, most existing studies focus primarily on leveraging meta-heuristic algorithms to improve the performance of machine learning models, whereas the application of machine learning concepts to enhance meta-heuristic algorithms has received relatively less attention. To address this research gap, this study proposes two novel strategies inspired by the KNN algorithm, logistic regression, and the entropy-weighting method from information concept to enhance the performance of swarm intelligence algorithms. These strategies include a logistic-KNN-based inertia weight mechanism and an entropy-weighted crossover mutation strategy, designed to tackle common challenges in meta-heuristic algorithms such as susceptibility to local optima, insufficient convergence speed, reduced population diversity, and premature convergence.

To validate the effectiveness of the proposed strategies, they were applied to the Seahorse Optimization Algorithm (SHO) and evaluated comprehensively using the IEEE benchmark test functions. The experimental results demonstrate that the proposed strategies significantly enhance the optimization capabilities of SHO, confirming their potential and effectiveness in improving swarm intelligence algorithms.

This research aims to provide new insights into addressing common issues in swarm intelligence optimization, ultimately contributing to the development of efficient optimization methods.

Introduction to the seahorse optimization algorithm

The Seahorse Optimization Algorithm (SHO) simulates the behavior of seahorses in nature, including movement, predation, and reproduction, to balance between global exploration and local exploitation in optimization tasks. The following sections provide a detailed description of the mathematical models and parameters associated with each behavior.

Initialization strategy

In SHO, each seahorse represents a candidate solution in the search space. The population matrix, denoted as Seahorses, is initialized as follows in Eq. (1):

(1) Seahorses=[x11⋯xDim,1⋮⋱⋮x1,pop⋯xDim,pop]

where Dim represents the dimension of the search space, and pop denotes the population size.

Each individual seahorse Xi is initialized within the boundary range [LB, UB] as shown in Eq. (2):

(2) xji=rand×(UBj−LBj)+LBj

where rand is a random number in the range [0,1], and LBj and UBj are the lower and upper bounds for the j-th variable, respectively.

The individual with the smallest fitness value in the initial population is selected as the elite individual Xelite, as shown in Eq. (3):

(3) Xelite=arg⁡minf(Xi)

where f(⋅) denotes the objective function of the optimization problem.

Movement behavior

SHO employs two movement patterns to balance exploration and exploitation: spiral motion and Brownian motion.

Spiral motion (local exploitation)

When a seahorse is influenced by ocean eddies, it moves along a spiral path toward the elite individual Xelite, as defined in Eq. (4):

(4) Xnew(1)(t+1)=Xi(t)+Levy(λ)×(Xelite(t)−Xi(t))×(x×y×z+Xelite(t))

where ‘t’ represents the current iteration count of the algorithm, and x, y, and z are given by Eq. (5):

(5) {x=ρ×cos⁡(θ)y=ρ×sin⁡(θ)z=ρ×θ

The parameter ρ is defined in Eq. (6):

(6) ρ=u×eθv

where u=0.05, v=0.05, and θ is a random value in the range [0,2π].

The Lévy flight step size Levy(λ) is computed as in Eq. (7):

(7) Levy(z)=s⋅w⋅σ/|k|1/λ

where s=0.01, and w and k are random numbers in [0,1]. The parameter σ is defined as in Eq. (8):

(8) σ=Γ(1+λ)⋅sin⁡(πλ/2)Γ((1+λ)/2)⋅λ⋅2(λ−1)/2

In this study, λ is set to 1.5.

Brownian motion (global exploration)

If the seahorse moves in response to ocean waves, it follows a Brownian path as defined in Eq. (9):

(9) Xnew(1)(t+1)=Xi(t)+rand×l×βt×(Xi(t)−βt×Xelite)

where l=0.05, and βt is a step-size coefficient following a standard normal distribution, as given by Eq. (10):

(10) βt=12πexp⁡(−x22)

Predation behavior

The predation behavior simulates a 90% probability of successful predation.

Predation success (90% probability)

When predation is successful, the seahorse moves closer to the elite individual, as shown in Eq. (11):

(11) Xnew(2)(t+1)=α×(Xelite−rand×Xnew(1)(t))+(1−α)×Xelite

Predation failure (10% Probability)

If predation fails, the seahorse explores a larger space as defined in Eq. (12):

(12) Xnew(2)(t+1)=(1−α)×(Xnew(1)(t)−rand×Xelite)+α×Xnew(1)(t)

The parameter α gradually decreases with iterations, calculated by Eq. (13):

(13) α=1−tT

where T is the maximum number of iterations.

Reproduction behavior

In SHO, reproduction generates new individuals that inherit traits from parent individuals to enhance population diversity.

Individual grouping

The population is sorted by fitness, with the top half as fathers and the bottom half as mothers, as shown in Eqs. (14), (15):

(14) Fathers={X(i)∣i=1,…,n2},

(15) Mothers={X(i)∣i=n2+1,…,n}.

where X(i) represents all individuals sorted in ascending order by fitness.

Offspring generation

Each pair of parents produces a new individual as shown in Eq. (16):

(16) Xoffspringi=r3×Xfatheri+(1−r3)×Xmotheri

where r3 is a random number in the range [0,1], and Xfatheri and Xmotheri are randomly selected father and mother individuals, respectively.

Related work

Inspiration

The logistic function, as a classical nonlinear model, is well-known in statistics and machine learning for its smooth S-shaped curve. Inspired by the characteristics of the Logistic function, the design of inertia weight in swarm intelligence algorithms can leverage its nonlinear mapping properties to achieve adaptive weight adjustment by balancing the exploration and exploitation behaviors of individuals.

The KNN algorithm is widely used in machine learning for classification and regression tasks. Its core concept involves calculating the distance between a target point and training data points to identify the nearest neighbors, and then predicting the target point’s attributes based on those neighbors’ characteristics. Drawing inspiration from this idea, the distance measurement mechanism of KNN can be introduced into swarm intelligence algorithms to strengthen the correlation between individuals and the global optimum.

By integrating the logistic function with the KNN method, an intelligent and adaptive inertia weight adjustment mechanism can be developed. This mechanism allows the dynamic adjustment of weights to more accurately reflect the distance between individuals and the global optimum, as well as the overall distribution characteristics of the population. Specifically, the KNN method serves as a distance-based quantitative tool to evaluate the relative relationship between individuals and their neighborhood or the global optimum. Meanwhile, the nonlinear mapping properties of the logistic function enable smooth and flexible adjustments of inertia weights based on the degree of aggregation within the population, thereby optimizing the weight adjustment process.

This approach allows the inertia weight to intelligently adapt to the dynamic changes during algorithm iterations, enabling real-time adjustments to balance exploration and exploitation effectively. The proposed strategy demonstrates significant advantages in enhancing the performance of swarm intelligence algorithms, particularly in accelerating convergence and improving global optimization capabilities. Moreover, it reduces the risk of the algorithm becoming trapped in local optima, further improving the robustness and efficiency of global search processes.

Furthermore, it can be envisioned that individuals within the population carry optimization information (such as lower or higher fitness values). Based on this information, the fitness level of the region where an individual is located can be roughly estimated. In traditional entropy weighting methods, “entropy” serves as an indicator of information richness, measuring the amount of information contained in the data. Therefore, in this study, the fitness value of each individual is treated as the “entropy” it contains, which is then used to guide individuals toward regions with superior information, thereby further enhancing the optimization process.

Logistic-KNN inertia weight

In swarm intelligence algorithms, the strategy of inertia weight plays a crucial role in algorithm performance. Typically, an ideal inertia weight should satisfy the following properties: Decreasing trend: The inertia weight should gradually decrease with the increase of iterations, ensuring that the search strategy shifts from global to local convergence.

Large initial value: A larger initial weight is favorable in the early stages of the algorithm, enhancing the population’s ability for global exploration and promoting diversity.

Small final value: A smaller weight toward the end of iterations supports a more detailed local search.

An appropriate inertia weight strategy can effectively guide the population to balance exploration and exploitation in the solution space, thereby enhancing the algorithm’s ability to navigate global and local search areas. However, traditional linearly decreasing inertia weights are overly rigid, potentially leading to insufficient flexibility in guiding strategies or premature convergence in complex optimization tasks. Consequently, researchers have proposed nonlinear inertia weight strategies, with the logistic function-based nonlinear weight being among the most widely applied. In such functions, the weight decreases gradually in the early stage, allows adjustable convergence in the mid-stage, and stabilizes in the later stages. Due to its favorable mathematical properties, this approach has gained broad acceptance.

However, such inertia weight strategies are designed based on the entire population, and thus lack precision in adjusting for individual differences. Consequently, traditional weight strategies fail to provide adaptive guidance for certain outlier individuals within the population. To address this issue, this study proposes a learning strategy inspired by the KNN algorithm, termed the distance-based learning strategy (KNN-Weight), integrating it into the nonlinear inertia weight design. This approach enables adaptive inertia guidance for each individual. The specific model is as follows:

In the proposed algorithm, the inertia weight lki(t) is dynamically adjusted based on distance, while leveraging the nonlinear characteristics of the Logistic function to enhance adaptive control for individuals. The mathematical model for inertia weight is shown in Eq. (17):

(17) lki(t)=wmin+(wmax−wmin)1+e−k(di(t)−dmean)

where wmin and wmax are two hyperparameters used to adjust the weight magnitude, respectively; k is a parameter that controls the rate of change in inertia weight, and di(t) represents the distance between individual i and the global best solution at iteration t. The formula for di(t) has three calculation methods,as shown in Eq. (18):

(18) {diEuclidean(t)=∥Xi(t)−Xbest(t)∥2diMahalanobis(t)=(Xi(t)−Xbest(t)TS−1(Xi(t)−Xbest(t))diMinkowski(t)=(∑j=1n|xi(j)(t)−xbest(j)(t)|p)1pp≥2

where Xi(t) denotes the position of individual i at iteration t, S−1 represents the inverse of the covariance matrix of the current iteration individual’s fitness value, which measures the correlation and variance between the individual’s dimensions (Euclidean distance is used by default), and Xbest(t) denotes the position of the global best solution at iteration t. The mean distance of the population dmean is defined as shown in Eq. (19):

(19) dmean=1N∑i=1Ndi(t)

By applying this inertia weight strategy to each individual, the algorithm can adjust different search strategies based on the population’s dispersion. When di(t)>dmean, the inertia weight lki(t) is larger, allowing individuals to perform broader exploration. Conversely, when di(t)<dmean, a smaller inertia weight lki(t) encourages more detailed local search. The logistic function’s favorable nonlinear properties allow the weight to smoothly control the rate of descent, providing adaptive flexibility. Increasing the parameter k enhances the inertia weight of individuals distant from the global best solution, further optimizing the algorithm’s convergence behavior and global exploration capabilities. To more intuitively illustrate the variation pattern of inertia weights, this study has generated function plots of inertia weights under different values of k, as shown in Fig. 1. Additionally, the complete process of the logistic-KNN inertia weight strategy is depicted in Fig. 2.

Figure 1 Diagram of logistic-KNN-Inertia weight (x-axis represents di, y-axis represents the corresponding value).

Figure 2 Diagram of the logistic-KNN-inertia weight strategy model.

Entropy crossover strategy

In group intelligence algorithms, the ‘Uniform Crossover Mutation’ strategy was first introduced in genetic algorithms (GAs). This strategy is described by a mathematical optimization model that captures the recombination and random mutation of individuals to generate a new generation. The classic mathematical model is shown in Eqs. (20), (21):

(20) CA=w⋅PA+(1−w)⋅PB,

(21) CB=(1−w)⋅PA+w⋅PB

where, w is a randomly generated weighting factor within the interval [0, 1] or ome random numbers between [0, 1] (e.g., standard normal distribution), and P and C represent the positional information of the parent and offspring, respectively. However, this random weighting factor fails to effectively capture critical information from both the father and mother, which limits the ability of new solutions to inherit superior traits from the parent generation. To address the limitations of random weighting in representing information, this study proposes an entropy-weighted crossover mutation strategy. By introducing the entropy-weighting method to determine the random parameter, the proposed strategy better incorporates the genetic information of the previous generation, thereby enhancing the algorithm’s performance in global optimization. 1. Extraction of parental fitness values

First, the population is sorted by fitness, with the top half designated as fathers and the bottom half as mothers. The fitness values are represented as shown in Eqs. (22), (23):

(22) Ffather={f1,f2,⋯,fn/2},

(23) Fmother={fn/2+1,fn/2+2,⋯,fn}

where Ffather and Fmother represent the sets of fitness values for fathers and mothers, respectively. 2. Normalization of fitness values

The fitness values of both fathers and mothers are normalized to facilitate subsequent weight calculations. The normalization formula is shown in Eqs. (24), (25):

(24) ffather_norm=ffather−min(Ffather)max(Ffather)−min(Ffather)+ε,

(25) fmother_norm=fmother−min(Fmother)max(Fmother)−min(Fmother)+ε,

where ε is a small constant to prevent division by zero. 3. Calculation of parental weights

After normalization, the fitness values are used to measure the extent to which an individual contains ‘better’ information. Individuals with smaller fitness values carry more valuable information and, from an information-theoretic perspective, have higher entropy. Conversely, individuals with larger fitness values are considered to contain less valuable information, resulting in lower entropy. The specific calculation is as shown in Eq. (26):

(26) wfather=1−ffather_norm,wmother=1−fmother_norm

4. Normalization of weights

Finally, the weights of fathers and mothers are normalized so that their sum equals 1, as shown in Eqs. (27), (28):

(27) wtotal=wfather+wmother

(28) w^father=wfatherwtotal,w^mother=wmotherwtotal

After normalization, the sum of the weights for fathers and mothers should equal to 1 as Eq. (29):

(29) w^father+w^mother=1

Through this fitness-weighted combination approach, individuals with higher fitness contribute more to the genes of the offspring, enhancing the balance between global exploration and local exploitation in the algorithm. This improved crossover and mutation strategy can more effectively guide the algorithm toward the global optimum, ensuring effective transmission of fitness information while retaining randomness.

Entropy-distance-SHO

To evaluate the effectiveness of our improved strategies, this study applies these two enhancements to the SHO algorithm. The improved algorithm is named Entropy-Distance-SHO. The strategy improved through Logistic-KNN-Weight is shown in Eq. (30):

(30) Xnew1(t+1)={lk(i)×Xi(t)+Levy(λ)((Xelite(t)−Xi(t))×xyz+Xelite(t)),r1>0lk(i)×Xi(t)+rand×l×βt×(Xi(t)−βt×Xelite),r1≤0

The crossover mutation strategy modified by the entropy-weighting method is shown in Eq. (31):

(31) Xioffspring=w^fatherXifather+w^motherXimother

where w^father represents the information weight from the father, and w^mother represents the information weight from the mother. The principle of this strategy is intuitively demonstrated in Fig. 3. The pseudocode of the algorithm is shown in Algorithm 1.

Figure 3 Diagram of the entropy weight cross-mutation strategy.

Algorithm 1 Entropy-Distance Seahorse Optimization (EDSHO) pseudocode.

 1: Initialize population X	
 2: Configure the EDSHO parameters ( lkmin, lkmax, etc.)	
 3: Calculate the fitness value of each individual	
 4: Determine the best individual Xelite	
 5: while t<T do	
 6:    if r1=rand(0,1)>0 then	
 7:      Update positions of the sea horses using Eq. (30)	
 8:   else	
 9:       Update positions of the population using Eq. (9)	
10:    end if	
11:    Update positions of the sea horses using Eqs. (11), (12)	
12:    Handle variables out of bounds	
13:    Calculate the fitness value of each individual	
14:    Select mothers and fathers using Eqs. (22), (23):	
15:    Breed offspring using Eq. (31)	
16:    Handle variables out of bounds	
17:    Calculate the fitness value of offspring	
18:    Select the next iteration population from the offspring and parents ranked top in fitness values	
19:    Update the best individual Xelite position	
20:     t=t+1	
21: end while	

Time complexity analysis

The time complexity of the algorithm is primarily determined by initialization, fitness evaluation, and the main loop, which includes inertia weight updates, predation behavior, reproduction, and fitness sorting. In each iteration, the complexity of inertia weight updates and predation behavior is O(pop×Dim), while reproduction and fitness sorting have complexities of O(pop×Dim) and O(poplog⁡pop), respectively. Therefore, the time complexity for a single iteration can be expressed as O(pop×Dim+poplog⁡pop). Taking into account the maximum number of iterations Max_iter, the overall time complexity of the algorithm is O(Max_iter×pop×Dim). When the problem dimension Dim is large, the impact of poplog⁡pop becomes negligible compared to pop×Dim, making the time complexity predominantly governed by O(Max_iter×pop×Dim). However, the time complexity of the improved algorithm has not increased, demonstrating the superiority of the proposed improvement strategies.

Experiment

Experiment overview

This study verifies the effectiveness of the proposed Entropy-Distance Seahorse Optimization (EDSHO) algorithm through three experimental groups: 1. Qualitative Analysis of the Algorithm

2. Benchmark Testing

3. Statistical Analysis

4. Ablation Experiment

The study uses 31 benchmark functions from the CEC2005 and CEC2021 suite to evaluate the EDSHO algorithm’s performance. These well-known optimization problems are widely used in various real-world applications, and the structure of these functions is detailed in Table 2. To ensure fairness, all algorithms are configured with a population size of 100 and 1,000 iterations. All experiments are conducted on the MATLAB 2024a platform, using a PC with an AMD Ryzen 7 6800H processor with Radeon Graphics, clocked at 3.20 GHz.

Table 2 CEC2005-CEC2021 test functions (F19–F27 contain bias terms).

Test function	Dim	Range	Fmin	
f1(x)=∑i=1nxi2	30	[−100,100]d	0	
f2(x)=∑i=1n|xi|+∏i=1n|xi|	30	[−10,10]d	0	
f3(x)=∑i=1n(∑j=1ixj)2	30	[−100,100]d	0	
f4(x)=max1≤i≤n|xi|	30	[−100,100]d	0	
f5(x)=∑i=1n−1[100(xi+1−xi2)2 +(xi−1)2]	30	[−30,30]d	0	
f6(x)=∑i=1n(⌊xi+0.5⌋)2	30	[−100,100]d	0	
f7(x)=∑i=1nixi4+random(0,1)	30	[−1.28,1.28]d	0	
f8(x)=∑i=1n−xisin⁡(|xi|)	30	[−500,500]d	−12,569.5	
f9(x)=∑i=1n|xi|2−10cos⁡(2πxi)+10	30	[−5.12,5.12]d	0	
f10(x)=−20exp⁡(−0.21n∑i=1nxi2) −exp⁡(1n∑i=1ncos2πxi)+20+e	30	[−32,32]d	0	
f11(x)=14000∑i=1nxi2−∏i=1ncos(xii)+1	30	[−600,600]d	0	
f12(x)=πn[10sin2(πy1)+∑i=1n−1(yi−1)2⋅ [1+10sin2(πyi+1)]+(yn−1)2] +∑i=1nu(xi,10,100,4)	30	[−50,50]d	0	
yi=1+xi+14				
f13(x)=0.1[sin2(3πx1)+∑i=1n−1(xi−1)2⋅ [1+sin2(3πxi+1)]+(xn−1)2[1+sin2(2πxn)]] +∑i=1nu(xi,5,100,4)	30	[−50,50]d	0	
f14(x)=[1500+∑j=1251j+∑i=1n(xi−aij)6]−1	2	[−65.536,65.536]d	1	
f15(x)=∑i=1nxi4−16xi2+5xi	4	[−5,5]d	−10.1532	
f16(x)=∑i=1nixi2+10(1−18π2)cos⁡x1+10	2	[−5,10]×[0,15]	0.398	
f17(x)=[1+(x1+x2+1)2(19−14x1+3x12 −14x2+6x1x2+3x22)]⋅[30+(2x1−3x2)2 (18−32x1+12x12+48x2−36x1x2+27x22)]	2	[−2,2]d	3	
f18(x)=−∑i=14ciexp⁡(−∑j=1naij(xj−pij)2)	4	[0,1]d	−3.86	
f19(x) = Shifted and Rotated Bent Cigar Function (CEC 2017[4] F1)	20	[−100,100]d	100	
f20(x) = Shifted and Rotated Schwefel’s Function (CEC 2014[3] F11)	20	[−100,100]d	1,100	
f21(x) = Shifted and Rotated Lunacek bi-Rastrigin Function (CEC 2017[4] F7)	20	[−100,100]d	700	
f22(x) = Expanded Rosenbrock’s plus Griewangk’s Function (CEC 2017[4])	20	[−100,100]d	1,900	
f23(x) = Hybrid Function 1 (N = 3) (CEC 2014[3] F17)	20	[−100,100]d	1,700	
f24(x) = Hybrid Function 2 (N = 4) (CEC 2017[4] F16)	20	[−100,100]d	1,600	
f25(x) = Hybrid Function 3 (N = 5) (CEC 2014[3] F21)	20	[−100,100]d	2,100	
f26(x) Composition Function 1 (N = 3) (CEC 2017[4] F22)	20	[−100,=100]d	2,200	
f27(x) = Composition Function 3 (N = 5) (CEC 2017[4] F25)	20	[−100,100]d	2,500	
f28(x)=−∑i=14ciexp⁡(−∑j=1naij(xj−pij)2)	4	[0,1]d	−3.32	
f29(x)=∑i=1n|xisin⁡(xi)+0.1xi|	4	[0,10]d	−10	
f30(x)=∑i=1n(xi−ai)2	4	[0,10]d	−10	
f31(x)=∑i=1n(xi−ai)T(xi−ci)−1	4	[0,10]d	−10	

Qualitative analysis

This study performs a detailed analysis of the EDSHO algorithm on high-dimensional unimodal, multimodal, and fixed-dimensional multimodal functions. The population size is set to 30, and the number of iterations is 300, with 30 independent runs on each test function. The results are shown in Figs. 4–6, with the analysis including the following aspects: 1. Function space search: The first column presents the distribution of the algorithm’s search in the function space.

2. Search history: The second column shows the search trajectory on the first and second dimensions, illustrating the population distribution within the search space. It is observed that on unimodal functions, the algorithm’s distribution is more concentrated, with more consistent optimal values found. In contrast, in high-dimensional function spaces, the search distribution is more scattered, and the optimal values found are similarly dispersed.

3. Mean fitness values: The third column displays the curve of the population’s mean fitness values. It is evident that the algorithm converges quickly after only a few iterations, reflecting its rapid convergence speed.

4. Optimal individual search trajectory: The fourth column records the search trajectory of the optimal individual on the first dimension. On unimodal functions, the curve shows larger oscillations in the initial stages and stabilizes in the later stages. For multimodal functions, there are also significant fluctuations initially, followed by small, persistent fluctuations.

5. Convergence curve: The fifth column is the convergence curve of the algorithm, clearly showing that the algorithm has high convergence speed and strong exploration capabilities.

Figure 4 High dimension unimodal functions.

Figure 5 High dimension multimodal functions.

Figure 6 Fixed-dimension multimodal functions.

Benchmark testing

This study compares the EDSHO algorithm against eight well-known algorithms recently proposed in the literature on the CEC2005 benchmark functions, including: SHO, GWO, nutcracker optimization algorithm (NOA) (Abdel-Basset et al., 2023), dung beetle optimizer (DBO) (Xue & Shen, 2023), snake optimizer (SO) (Hashim & Hussien, 2022), pigeon-inspired optimization (PIO) (Duan & Qiao, 2014), whale optimization algorithm (WOA) (Mirjalili & Lewis, 2016), and the seagule optimization algorithm (SOA) (Dhiman & Kumar, 2019). During testing, each algorithm is run 30 times independently on each test function, and the reported metrics include mean fitness value, standard deviation, best value, and worst value. Detailed results are shown in Tables 3–5, and Figs. 7–10 provides an intuitive visualization of algorithms performance.

Table 3 Fitness values for algorithm across function F1 to F11 (Mean, Std, Best, Worst).

Function	Metric	Algorithm	
		EDSHO	SHO	SOA	WOA	PIO	SO	DBO	NOA	GWO	
F1	Mean	0.00×100	5.41×10−305	1.70×10−33	6.21×10−190	6.45×10−216	2.37×10−195	8.36×10−268	5.10×104	4.10×10−85	
	Std	0.00×100	0.00×100	3.62×10−33	0.00×100	0.00×100	0.00×100	0.00×100	6.78×103	6.28×10−85	
	Best	0.00×100	5.04×10−313	1.68×10−35	1.78×10−205	2.43×10−231	2.01×10−200	0.00×100	3.17×104	2.49×10−87	
	Worst	0.00×100	1.08×10−303	1.64×10−32	1.24×10−188	6.41×10−215	2.09×10−194	1.67×10−266	5.95×104	2.21×10−84	
F2	Mean	2.52×10−292	5.96×10−169	1.15×10−20	6.96×10−114	7.53×10−107	7.17×10−100	2.07×10−140	7.57×109	3.83×10−49	
	Std	0.00×100	0.00×100	1.55×10−20	2.36×10−113	2.76×10−106	1.02×10−99	8.98×10−140	3.09×1010	3.45×10−49	
	Best	3.98×10−301	4.32×10−174	8.47×10−22	3.25×10−123	3.64×10−117	8.34×10−102	4.94×10−181	1.37×106	5.53×10−50	
	Worst	3.02×10−291	4.63×10−168	5.86×10−20	1.04×10−112	1.22×10−105	3.59×10−99	4.02×10−139	1.39×1011	1.35×10−48	
F3	Mean	0.00×100	5.55×10−218	2.22×10−19	4.73×103	3.62×10−214	2.99×10−130	1.80×10−93	7.30×104	4.96×10−27	
	Std	0.00×100	0.00×100	3.67×10−19	4.04×103	0.00×100	1.33×10−129	8.06×10−93	1.32×104	9.40×10−27	
	Best	0.00×100	2.07×10−228	1.46×10−23	9.43×101	2.53×10−221	1.22×10−144	0.00×100	4.38×104	6.58×10−31	
	Worst	0.00×100	9.93×10−217	1.54×10−18	1.27×104	6.62×10−213	5.97×10−129	3.60×10−92	1.05×105	3.82×10−26	
F4	Mean	2.67×10−274	2.37×10−119	1.60×10−10	2.19×101	1.05×10−107	6.31×10−89	3.60×10−96	8.00×101	1.86×10−21	
	Std	0.00×100	8.42×10−119	4.12×10−10	2.11×101	3.93×10−107	2.22×10−88	1.61×10−95	3.09×100	3.82×10−21	
	Best	9.89×10−288	4.12×10−124	5.66×10−13	5.50×10−7	8.28×10−113	1.13×10−91	2.96×10−178	7.05×101	1.86×10−23	
	Worst	5.31×10−273	3.78×10−118	1.74×10−9	5.48×101	1.76×10−106	1.00×10−87	7.20×10−95	8.33×101	1.51×10−20	
F5	Mean	2.09×101	2.74×101	2.76×101	2.59×101	2.89×101	4.36×101	2.32×101	1.54×108	2.65×101	
	Std	1.12×101	6.57×10−1	5.40×10−1	1.98×10−1	7.73×10−2	4.60×101	2.72×10−1	2.91×107	7.19×10−1	
	Best	9.84×10−2	2.63×101	2.71×101	2.57×101	2.88×101	2.87×101	2.27×101	9.47×107	2.52×101	
	Worst	2.80×101	2.88×101	2.88×101	2.65×101	2.90×101	2.00×102	2.38×101	1.88×108	2.87×101	
F6	Mean	6.31×10−2	2.20×100	2.40×100	3.40×10−4	3.81×100	1.26×100	3.73×10−25	5.04×104	1.38×10−1	
	Std	1.79×10−1	4.41×10−1	5.81×10−1	1.42×10−4	1.87×100	9.18×10−1	4.73×10−25	4.80×103	1.91×10−1	
	Best	2.37×10−4	1.53×100	1.25×100	1.36×10−4	9.47×10−1	7.24×10−2	1.18×10−26	3.99×104	4.59×10−6	
	Worst	7.50×10−1	3.25×100	3.49×100	6.78×10−4	6.98×100	3.57×100	1.92×10−24	5.95×104	7.53×10−1	
F7	Mean	1.50×10−5	1.60×10−5	2.84×10−4	5.91×10−4	5.39×10−5	3.47×10−5	1.16×10−3	7.04×101	2.85×10−4	
	Std	1.34×10−5	1.66×10−5	1.86×10−4	6.44×10−4	4.77×10−5	2.86×10−5	8.33×10−4	1.02×101	1.63×10−4	
	Best	2.01×10−6	7.34×10−7	6.03×10−5	1.33×10−5	4.92×10−6	3.05×10−6	1.66×10−4	4.98×101	7.84×10−5	
	Worst	6.17×10−5	7.76×10−5	7.43×10−4	1.86×10−3	1.68×10−4	1.18×10−4	3.51×10−3	8.73×101	6.86×10−4	
F8	Mean	−4.18×104	−1.39×104	−1.31×104	−4.08×104	−1.29×104	−1.82×104	−3.06×104	−1.81×104	−1.75×104	
	Std	1.31×102	4.97×102	2.31×103	1.32×103	1.08×103	7.46×10−12	6.19×103	7.46×10−12	1.85×103	
	Best	−4.19×104	−1.51×104	−1.70×104	−4.19×104	−1.52×104	−1.81×104	−4.16×104	−1.81×104	−2.21×104	
	Worst	−4.14×104	−1.32×104	−1.02×104	−3.72×104	−1.12×104	−1.81×104	−2.29×104	−1.81×104	−1.40×104	
F9	Mean	0.00×100	0.00×100	0.00×100	2.84×10−15	0.00×100	4.68×101	3.00×101	3.77×102	9.07×10−1	
	Std	0.00×100	0.00×100	0.00×100	1.27×10−14	0.00×100	1.22×101	5.47×101	1.85×101	2.88×100	
	Best	0.00×100	0.00×100	0.00×100	0.00×100	0.00×100	2.80×101	0.00×100	3.33×102	0.00×100	
	Worst	0.00×100	0.00×100	0.00×100	5.68×10−14	0.00×100	6.67×101	1.56×102	4.03×102	1.12×101	
F10	Mean	4.44×10−16	3.82×10−15	1.90×101	3.11×10−15	4.44×10−16	2.00×101	4.44×10−16	2.00×101	9.86×10−15	
	Std	0.00×100	7.94×10−16	4.46×100	2.79×10−15	0.00×100	2.03×10−3	0.00×100	7.29×10−15	2.89×10−15	
	Best	4.44×10−16	4.44×10−16	2.18×10−14	4.44×10−16	4.44×10−16	2.00×101	4.44×10−16	2.00×101	7.55×10−15	
	Worst	4.44×10−16	4.00×10−15	2.00×101	7.55×10−15	4.44×10−16	2.00×101	4.44×10−16	2.00×101	1.47×10−14	
F11	Mean	0.00×100	0.00×100	1.15×10−3	0.00×100	0.00×100	9.96×10−2	0.00×100	4.69×102	5.12×10−4	
	Std	0.00×100	0.00×100	5.14×10−3	0.00×100	0.00×100	1.69×10−1	0.00×100	3.05×101	2.29×10−3	
	Best	0.00×100	0.00×100	0.00×100	0.00×100	0.00×100	0.00×100	0.00×100	4.05×102	0.00×100	
	Worst	0.00×100	0.00×100	2.30×10−2	0.00×100	0.00×100	4.45×10−1	0.00×100	5.25×102	1.02×10−2	

Table 4 Fitness values for algorithm across function F12 to F22 (Mean, Std, Best, Worst).

Function	Metric	Algorithm	
		EDSHO	SHO	SOA	WOA	PIO	SO	DBO	NOA	GWO	
F12	Mean	3.61×10−4	1.20×10−1	1.92×10−1	5.56×10−5	8.52×10−1	1.81×100	2.94×10−13	2.83×108	1.40×10−2	
	Std	1.46×10−3	4.15×10−2	6.38×10−2	3.49×10−5	4.11×10−1	2.01×100	1.31×10−12	5.80×107	1.17×10−2	
	Best	8.49×10−7	4.82×10−2	1.07×10−1	1.45×10−5	1.42×10−1	3.51×10−1	4.18×10−29	1.78×108	2.92×10−7	
	Worst	6.56×10−3	2.03×10−1	3.38×10−1	1.53×10−4	1.39×100	6.69×100	5.88×10−12	4.03×108	3.31×10−2	
F13	Mean	5.55×10−4	1.43×100	1.67×100	5.80×10−3	1.26×100	2.60×100	5.90×10−2	6.12×108	1.29×10−1	
	Std	2.93×10−4	3.12×10−1	1.41×10−1	1.09×10−2	5.72×10−1	5.67×10−1	6.68×10−2	1.15×108	9.78×10−2	
	Best	1.51×10−4	1.02×100	1.24×100	4.72×10−4	7.00×10−1	1.41×100	3.91×10−26	3.76×108	6.07×10−6	
	Worst	1.03×10−3	2.20×100	1.86×100	4.69×10−2	2.67×100	3.00×100	2.07×10−1	8.06×108	3.00×10−1	
F14	Mean	9.98×10−1	2.57×100	9.98×10−1	9.98×10−1	9.98×10−1	9.98×10−1	9.98×10−1	4.83×100	2.67×100	
	Std	3.78×10−16	2.95×100	1.69×10−9	8.48×10−13	9.23×10−8	0.00×100	0.00×100	2.24×100	2.92×100	
	Best	9.98×10−1	9.98×10−1	9.98×10−1	9.98×10−1	9.98×10−1	9.98×10−1	9.98×10−1	1.07×100	9.98×10−1	
	Worst	9.98×10−1	1.08×101	9.98×10−1	9.98×10−1	9.98×10−1	9.98×10−1	9.98×10−1	9.59×100	1.08×101	
F15	Mean	3.08×10−4	4.27×10−4	1.23×10−3	6.62×10−4	5.44×10−4	1.88×10−3	5.24×10−4	1.36×10−2	3.99×10−4	
	Std	8.38×10−8	3.63×10−4	1.11×10−5	4.04×10−4	1.35×10−4	4.37×10−3	2.76×10−4	9.24×10−3	2.82×10−4	
	Best	3.07×10−4	3.08×10−4	1.22×10−3	3.08×10−4	3.77×10−4	3.08×10−4	3.07×10−4	2.13×10−3	3.07×10−4	
	Worst	3.08×10−4	1.70×10−3	1.27×10−3	1.24×10−3	8.75×10−4	2.04×10−2	1.22×10−3	4.54×10−2	1.22×10−3	
F16	Mean	4.01×10−1	3.98×10−1	3.98×10−1	3.98×10−1	3.98×10−1	3.98×10−1	3.98×10−1	4.68×10−1	3.98×10−1	
	Std	5.08×10−3	8.42×10−5	2.79×10−5	4.61×10−9	1.61×10−4	0.00×100	0.00×100	6.56×10−2	3.93×10−5	
	Best	3.98×10−1	3.98×10−1	3.98×10−1	3.98×10−1	3.98×10−1	3.98×10−1	3.98×10−1	3.99×10−1	3.98×10−1	
	Worst	4.19×10−1	3.98×10−1	3.98×10−1	3.98×10−1	3.98×10−1	3.98×10−1	3.98×10−1	5.98×10−1	3.98×10−1	
F17	Mean	3.00×100	3.00×100	3.00×100	3.00×100	3.00×100	3.00×100	3.00×100	4.11×100	3.00×100	
	Std	2.27×10−8	4.18×10−12	1.63×10−6	1.67×10−7	1.72×10−3	7.62×10−16	1.06×10−15	1.63×100	5.88×10−7	
	Best	3.00×100	3.00×100	3.00×100	3.00×100	3.00×100	3.00×100	3.00×100	3.01×100	3.00×100	
	Worst	3.00×100	3.00×100	3.00×100	3.00×100	3.01×100	3.00×100	3.00×100	9.56×100	3.00×100	
F18	Mean	−3.85×100	−3.86×100	−3.85×100	−3.86×100	−3.86×100	−3.86×100	−3.86×100	−3.82×100	−3.86×100	
	Std	1.04×10−2	3.85×10−3	1.03×10−5	8.86×10−4	8.60×10−4	2.28×10−15	2.28×10−15	3.37×10−2	1.73×10−3	
	Best	−3.86×100	−3.86×100	−3.85×100	−3.86×100	−3.86×100	−3.86×100	−3.86×100	−3.86×100	−3.86×100	
	Worst	−3.82×100	−3.85×100	−3.85×100	−3.86×100	−3.86×100	−3.86×100	−3.86×100	−3.73×100	−3.86×100	
F19	Mean	0	0	8.2632×10−41	8.439×10−197	1.4006×10−166	3.2484×1010	2.9387×10−258	2.5484×1010	5.039×10−117	
	Std	0	0	1.8559×10−40	0	0	4.0803×109	0	3.7058×109	1.187×10−116	
	Best	0	0	2.2724×10−43	3.9194×10−207	−1.3507×10−165	2.2809×1010	0	1.8538×1010	5.4335×10−120	
	Worst	0	0	7.1746×10−40	1.2651×10−195	1.5303×10−165	3.9038×1010	4.4081×10−257	2.9607×1010	4.5544×10−116	
F20	Mean	0.35181	0	0	2.4253×10−13	9.8083×10−9	5,753.8	792.27	5,244.4	0.51212	
	Std	0.34086	0	0	6.4004×10−13	1.5182×10−8	450.63	714.89	322.76	1.9044	
	Best	0	0	0	0	−1.1548×10−8	4,686.1	0	4370.6	0	
	Worst	0.68456	0	0	1.819×10−12	2.843×10−8	6,624.2	1,765.6	5694	7.3908	
F21	Mean	2.672	0	13.231	0	−2.7201×10−16	1314.6	20.295	1137.3	76.492	
	Std	7.0515	0	51.242	0	6.6911×10−16	141.43	25.197	82.18	50.085	
	Best	0	0	0	0	−9.3544×10−16	925.46	0	993.92	0	
	Worst	20.178	0	198.46	0	1.0218×10−15	1493	61.283	1277.2	151.31	
F22	Mean	0	0	0	0.13556	1.1624×10−5	3.779×105	1.9682	1.8225×105	0.43506	
	Std	0	0	0	0.52504	8.9236×10−5	1.5558×105	1.2702	8.4300×104	0.57709	
	Best	0	0	0	0	−0.00017941	62651	0	55819	0	
	Worst	0	0	0	2.0335	0.00016585	6.5656×105	4.2395	3.9898×105	2.0289	

Table 5 Fitness values for algorithm across function F23 to F31 (Mean, Std, Best, Worst).

Function	Metric	Algorithm	
		EDSHO	SHO	SOA	WOA	PIO	SO	DBO	NOA	GWO	
F23	Mean	0	0	0.45188	4.2604×10−161	4.8196×10−90	8.8054×107	7.4333	3.0593×107	1.5338	
	Std	0	0	1.7501	1.6499×10−160	1.9842×10−89	4.9545×107	19.898	2.5427×107	3.7469	
	Best	0	0	2.4325×10−48	6.1177×10−203	−4.1463×10−90	5.679×106	0	5.9667×106	1.7563×10−133	
	Worst	0	0	6.7782	6.3907×10−160	7.644×10−89	1.4881×108	70.768	1.0254×108	14.291	
F24	Mean	1.9113×10−5	0.0026006	1.1889	0.10928	−5.398×10−16	2,086.6	42.622	1,524.1	1.6998	
	Std	2.7105×10−5	0.010047	1.9902	0.20146	1.185×10−15	285.07	52.841	288.44	2.0511	
	Best	2.9587×10−7	9.0584×10−8	2.8732×10−5	2.6066×10−5	−2.1978×10−15	1,711.2	0	912.01	0.049827	
	Worst	9.1011×10−5	0.038917	4.957	0.79059	1.0931×10−15	2636.7	166.86	1,921.2	7.0688	
F25	Mean	2.3569×10−5	3.6481×10−6	1.1259	0.030802	2.309×10−167	4.5221×107	15.681	8.5914×106	0.11995	
	Std	2.6854×10−5	4.857×10−6	1.9163	0.042324	0	2.4995×107	40.826	6.1265×106	0.13012	
	Best	3.5835×10−7	6.4502×10−13	2.6532×10−5	0.0012188	−1.3332×10−165	8.0353×106	−2.2204×10−16	8.4869×105	0.015988	
	Worst	8.171×10−5	1.7936×10−5	4.3247	0.16482	1.4483×10−165	9.9007×107	155.36	1.9835×107	0.47398	
F26	Mean	0	0	0	0	−8.65×10−11	5,652.3	6.1851	5,144.6	0	
	Std	0	0	0	0	9.891×10−10	288.71	23.955	314.4	0	
	Best	0	0	0	0	−1.6186×10−9	5,112.1	0	4,489.8	0	
	Worst	0	0	0	0	1.7071×10−9	6,173.9	92.777	5,523.6	0	
F27	Mean	0.00012062	0.00019848	39.135	0.097587	−1.8286×10−154	2,701.6	67.38	1,664.9	62.884	
	Std	0.00013623	0.00019863	43.684	0.040512	8.5794×10−154	482.78	17.981	347.16	13.827	
	Best	1.3934×10−6	1.2724×10−5	0.0014615	0.013993	−3.2641×10−153	2141.5	49.331	1,167.6	49.511	
	Worst	0.0004873	0.00080577	89.011	0.14831	3.6984×10−154	4,195.4	110.77	2,180.9	80.573	
F28	Mean	−2.74×100	−3.13×100	−3.04×100	−3.25×100	−3.14×100	−3.32×100	−3.25×100	−2.62×100	−3.26×100	
	Std	3.10×10−1	1.26×10−1	5.07×10−2	6.18×10−2	4.68×10−2	2.66×10−2	6.45×10−2	2.04×10−1	6.69×10−2	
	Best	−3.16×100	−3.32×100	−3.13×100	−3.32×100	−3.25×100	−3.32×100	−3.32×100	−3.00×100	−3.32×100	
	Worst	−2.01×100	−2.84×100	−3.02×100	−3.19×100	−3.05×100	−3.20×100	−3.14×100	−2.28×100	−3.14×100	
F29	Mean	−1.00×101	−6.62×100	−4.83×100	−1.02×101	−6.24×100	−8.66×100	−7.00×100	−1.46×100	−9.14×100	
	Std	1.88×10−1	2.79×100	4.02×100	2.62×10−4	1.71×100	2.70×100	2.70×100	5.41×10−1	2.08×100	
	Best	−1.02×101	−1.01×101	−1.01×101	−1.02×101	−9.49×100	−1.02×101	−1.02×101	−2.81×100	−1.02×101	
	Worst	−9.46×100	−8.82×10−1	−4.97×10−1	−1.02×101	−4.71×100	−2.63×100	−2.63×100	−7.76×10−1	−5.06×100	
F30	Mean	−1.02×101	−6.80×100	−7.69×100	−1.01×101	−5.38×100	−9.44×100	−6.16×100	−1.81×100	−1.01×101	
	Std	1.78×10−1	2.73×100	3.96×100	1.19×100	1.18×100	2.26×100	2.54×100	6.18×10−1	1.19×100	
	Best	−1.04×101	−1.04×101	−1.04×101	−1.04×101	−8.51×100	−1.04×101	−1.04×101	−3.39×100	−1.04×101	
	Worst	−9.87×100	−3.72×100	−9.08×10−1	−5.09×100	−4.72×100	−2.77×100	−2.77×100	−9.06×10−1	−5.09×100	
F31	Mean	−1.04×101	−6.20×100	−9.44×100	−1.03×101	−6.21×100	−1.01×101	−7.10×100	−1.80×100	−1.05×101	
	Std	1.12×10−1	2.20×100	2.21×100	1.21×100	1.68×100	1.51×100	2.75×100	4.00×10−1	1.60×10−4	
	Best	−1.05×101	−1.05×101	−1.05×101	−1.05×101	−1.01×101	−1.05×101	−1.05×101	−2.64×100	−1.05×101	
	Worst	−1.02×101	−5.13×100	−5.13×100	−5.13×100	−4.83×100	−3.84×100	−3.84×100	−1.33×100	−1.05×101	

Figure 7 Convergence curves of F1–F9.

Figure 8 Convergence curves of F10–F18.

Figure 9 Convergence curves of F19–F27.

Figure 10 Convergence curves of F28–F31.

The results indicate that EDSHO outperforms most other algorithms in terms of convergence speed and exploration capability on unimodal functions, with all metrics showing optimal values. On multimodal functions, EDSHO maintains stable performance across different types of optimization problems. For fixed-dimensional multimodal function optimization, EDSHO, WOA, and SO algorithms perform well; although EDSHO may slightly underperform in certain metrics, the differences remain minimal.

To further evaluate the performance of EDSHO, this study conducts the following statistical analyses.

Statistical analysis

This study conducted the Wilcoxon rank-Sum test on the results of each algorithm run independently. The results indicate that, on most test functions, the performance of each algorithm is significantly different from the others, with only a few cases showing no significant difference. In cases where the optimization results of two algorithms are identical, “N/A” is recorded. Detailed results are shown in Table 6.

Table 6 Wilcoxon rank-sum test P-values across functions F1 to F31.

Function	SHO	SOA	WOA	PIO	SO	DBO	NOA	GWO	
F1	8.01×10−9	8.01×10−9	8.01×10−9	8.01×10−9	8.01×10−9	3.31×10−6	8.01×10−9	8.01×10−9	
F2	6.80×10−8	6.80×10−8	6.80×10−8	6.80×10−8	6.80×10−8	6.80×10−8	6.80×10−8	6.80×10−8	
F3	8.01×10−9	8.01×10−9	8.01×10−9	8.01×10−9	8.01×10−9	1.10×10−6	8.01×10−9	8.01×10−9	
F4	6.80×10−8	6.80×10−8	6.80×10−8	6.80×10−8	6.80×10−8	6.80×10−8	6.80×10−8	6.80×10−8	
F5	3.51×10−1	6.04×10−3	9.05×10−3	6.80×10−8	6.80×10−8	7.11×10−3	6.80×10−8	7.64×10−2	
F6	6.80×10−8	6.80×10−8	9.28×10−5	6.80×10−8	2.96×10−7	6.80×10−8	6.80×10−8	3.65×10−1	
F7	1.00×100	7.90×10−8	1.05×10−6	1.61×10−4	1.06×10−2	6.80×10−8	6.80×10−8	6.80×10−8	
F8	6.80×10−8	6.80×10−8	1.90×10−1	6.80×10−8	2.96×10−8	9.17×10−8	8.01×10−9	6.80×10−8	
F9	N/A	N/A	3.42×10−1	N/A	8.01×10−9	9.58×10−3	8.01×10−9	8.06×10−2	
F10	3.13×10−9	8.01×10−9	1.55×10−4	N/A	2.67×10−9	N/A	4.68×10−10	5.00×10−9	
F11	N/A	3.42×10−1	N/A	N/A	9.58×10−3	N/A	8.01×10−9	3.42×10−1	
F12	6.80×10−8	6.80×10−8	1.20×10−1	6.80×10−8	6.80×10−8	6.80×10−8	6.80×10−8	1.48×10−3	
F13	6.80×10−8	6.80×10−8	1.35×10−3	6.80×10−8	6.80×10−8	7.10×10−3	6.80×10−8	1.23×10−3	
F14	4.15×10−4	6.05×10−8	6.05×10−8	6.05×10−8	6.88×10−9	6.88×10−9	6.05×10−8	6.05×10−8	
F15	3.99×10−6	6.80×10−8	2.22×10−7	6.80×10−8	2.21×10−7	6.36×10−1	6.80×10−8	3.29×10−5	
F16	3.05×10−4	5.12×10−3	9.17×10−8	1.48×10−1	8.01×10−9	8.01×10−9	2.36×10−6	1.10×10−5	
F17	2.96×10−7	1.05×10−6	9.09×10−2	6.80×10−8	2.95×10−8	2.95×10−8	6.80×10−8	1.58×10−6	
F18	3.50×10−6	1.23×10−3	1.23×10−7	3.07×10−6	8.01×10−9	8.01×10−9	1.79×10−4	1.23×10−7	
F19	N/A	6.8662×10−7	6.8662×10−7	0.75632	6.8662×10−7	2.7717×10−5	6.8662×10−7	6.8662×10−7	
F20	0.001523	0.001523	0.0094448	0.14405	2.9431×10−6	0.046242	2.9431×10−6	0.079012	
F21	0.16439	1	0.16439	0.0094312	1.2604×10−6	0.031667	1.2604×10−6	2.537×10−5	
F22	N/A	N/A	0.35065	0.32942	6.8662×10−7	2.7717×10−5	6.8662×10−7	8.6982×10−6	
F23	N/A	6.8662×10−7	6.8662×10−7	0.75632	6.8662×10−7	8.6982×10−6	6.8662×10−7	6.8662×10−7	
F24	0.18441	7.4772×10−6	7.4772×10−6	3.3918×10−6	3.3918×10−6	0.00013564	3.3918×10−6	3.3918×10−6	
F25	0.0018655	6.1516×10−6	3.3918×10−6	3.3918×10−6	3.3918×10−6	0.12486	3.3918×10−6	3.3918×10−6	
F26	N/A	N/A	N/A	0.75632	6.8662×10−7	0.35065	6.8662×10−7	N/A	
F27	0.13538	3.3918×10−6	3.3918×10−6	3.3918×10−6	3.3918×10−6	3.3918×10−6	3.3918×10−6	3.3918×10−6	
F28	8.60×10−6	1.16×10−4	6.80×10−8	2.22×10−7	1.13×10−8	4.56×10−8	3.37×10−1	6.80×10−8	
F29	1.95×10−3	1.78×10−3	6.80×10−8	7.90×10−8	2.84×10−1	2.84×10−1	6.80×10−8	1.23×10−3	
F30	1.06×10−2	9.03×10−1	6.67×10−6	6.80×10−8	9.53×10−4	1.36×10−3	6.80×10−8	2.04×10−5	
F31	3.29×10−5	7.15×10−1	6.67×10−6	6.80×10−8	1.18×10−5	3.08×10−2	6.80×10−8	3.42×10−7	

To further investigate the differences among the algorithms, a Friedman sum rank test was performed to rank their performance, as shown in Table 7, while Fig. 11 illustrates the relative performance levels of the algorithms. It is evident that EDSHO achieved excellent rankings in terms of average fitness, best fitness, and worst fitness values among all algorithms. Across all performance metrics, EDSHO consistently outperformed SHO, demonstrating the effectiveness of the improvements (significance level: α=0.05).

Table 7 Friedman sum rank test across different metrics.

Algorithm	Mean rank	Rank	Std rank	Rank	Best rank	Rank	Worst rank	Rank	
EDSHO	2.86	1	3.30	1	3.42	2	2.92	1	
WOA	3.77	2	4.17	3	4.33	5	3.80	2	
SHO	3.88	3	4.17	3	4.20	3	3.92	3	
PIO	4.28	4	3.91	2	4.23	4	4.22	4	
DBO	4.58	5	4.86	7	3.28	1	4.55	5	
GWO	4.88	6	5.47	8	4.66	6	4.81	6	
SOA	5.75	7	5.61	9	6.00	8	5.72	7	
SO	6.47	8	5.84	6	6.39	9	6.70	8	
NOA	8.55	9	7.67	10	8.48	10	8.36	9	

Figure 11 Algorithm ranking results.

Ablation experiment

To further evaluate the contribution of each component, this article adopts ablation experiments to investigate the specific impact of each component on the SHO algorithm. E_SHO represents the SHO algorithm with the entropy-based crossover mutation strategy, while D_SHO represents the SHO algorithm with the Logistic-KNN Inertia Weight strategy. The experiments were conducted on 18 CEC2005 benchmark test functions, and the reported evaluation metrics include mean, best, worst, and standard deviation (std). The results are presented in Tables 8, 9, while the convergence curves are shown in Figs. 12, 13.

Table 8 Ablation experiment results F1 to F8.

Function	Metric	SHO	E_SHO	D_SHO	EDSHO	
F1	Mean	9.31 × 10−306	0	0	0	
	Std	0	0	0	0	
	Best	1.96 × 10−314	0	0	0	
	Worst	1.02 × 10−304	0	0	0	
F2	Mean	1.47 × 10−168	1.84 × 10−197	3.47 × 10−288	1.38 × 10−290	
	Std	0	0	0	0	
	Best	6.19 × 10−172	5.27 × 10−204	3.33 × 10−295	6.88 × 10−306	
	Worst	2.10 × 10−167	1.84 × 10−196	4.74 × 10−287	2.07 × 10−289	
F3	Mean	4.49 × 10−217	7.69 × 10−294	0	0	
	Std	0	0	0	0	
	Best	1.32 × 10−226	6.11 × 10−305	0	0	
	Worst	6.26 × 10−216	1.01 × 10−292	0	0	
F4	Mean	1.61 × 10−118	4.08 × 10−160	1.98 × 10−277	6.16 × 10−276	
	Std	5.77 × 10−118	1.18 × 10−159	0	0	
	Best	8.69 × 10−122	6.65 × 10−166	1.30 × 10−284	7.93 × 10−287	
	Worst	2.25 × 10−117	4.55 × 10−159	2.87 × 10−276	9.20 × 10−275	
F5	Mean	2.77 × 101	2.44 × 101	2.06 × 101	1.31 × 101	
	Std	7.72 × 10−1	1.21 × 101	1.11 × 101	3.92 × 10−2	
	Best	2.63 × 101	3.37 × 10−1	1.50	2.86 × 101	
	Worst	2.88 × 101	5.35 × 101	2.80 × 101	1.75 × 101	
F6	Mean	2.44	4.11 × 10−1	5.58 × 10−4	6.18 × 10−4	
	Std	5.46 × 10−1	3.57 × 10−1	2.21 × 10−4	3.42 × 10−4	
	Best	1.26	6.59 × 10−2	2.53 × 10−4	2.10 × 10−4	
	Worst	3.25	1.11	8.91 × 10−4	1.62 × 10−3	
F7	Mean	1.42 × 10−5	1.62 × 10−5	1.07 × 10−5	1.66 × 10−5	
	Std	8.75 × 10−6	1.79 × 10−5	9.14 × 10−6	1.28 × 10−5	
	Best	1.89 × 10−6	1.59 × 10−6	6.89 × 10−7	3.04 × 10−7	
	Worst	2.79 × 10−5	7.28 × 10−5	3.40 × 10−5	4.22 × 10−5	
F8	Mean	−1.37 × 104	−4.16 × 104	−4.15 × 104	−4.17 × 104	
	Std	8.82 × 102	4.06 × 102	3.21 × 102	2.98 × 102	
	Best	−1.48 × 104	−4.19 × 104	−4.19 × 104	−4.19 × 104	
	Worst	−1.21 × 104	−4.08 × 104	−4.08 × 104	−4.09 × 104	
F9	Mean	0	0	0	0	
	Std	0	0	0	0	
	Best	0	0	0	0	
	Worst	0	0	0	0	

Table 9 Ablation experiment results F10 to F18.

Function	Metric	SHO	E_SHO	D_SHO	EDSHO	
F10	Mean	3.76 × 10−15	4.44 × 10−16	4.44 × 10−16	4.44 × 10−16	
	Std	9.17 × 10−16	0	0	0	
	Best	4.44 × 10−16	4.44 × 10−16	4.44 × 10−16	4.44 × 10−16	
	Worst	4.00 × 10−15	4.44 × 10−16	4.44 × 10−16	4.44 × 10−16	
F11	Mean	0	0	0	0	
	Std	0	0	0	0	
	Best	0	0	0	0	
	Worst	0	0	0	0	
F12	Mean	1.30 × 10−1	5.32 × 10−3	3.41 × 10−5	3.99 × 10−5	
	Std	6.29 × 10−2	6.40 × 10−3	2.23 × 10−5	1.99 × 10−5	
	Best	4.98 × 10−2	1.01 × 10−4	4.69 × 10−6	1.38 × 10−5	
	Worst	2.68 × 10−1	1.86 × 10−2	8.75 × 10−5	8.97 × 10−5	
F13	Mean	1.42	1.07 × 10−1	1.22 × 10−3	5.85 × 10−4	
	Std	2.84 × 10−1	9.52 × 10−2	2.86 × 10−3	2.88 × 10−4	
	Best	7.73 × 10−1	3.52 × 10−6	2.27 × 10−4	2.02 × 10−4	
	Worst	1.94	3.55 × 10−1	1.16 × 10−2	1.32 × 10−3	
F14	Mean	2.31	9.98 × 10−1	9.98 × 10−1	9.98 × 10−1	
	Std	2.52	7.96 × 10−16	3.80 × 10−16	2.78 × 10−16	
	Best	9.98 × 10−1	9.98 × 10−1	9.98 × 10−1	9.98 × 10−1	
	Worst	1.08 × 101	9.98 × 10−1	9.98 × 10−1	9.98 × 10−1	
F15	Mean	4.72 × 10−4	3.11 × 10−4	3.08 × 10−4	3.08 × 10−4	
	Std	4.02 × 10−4	3.45 × 10−6	4.39 × 10−8	3.00 × 10−7	
	Best	3.08 × 10−4	3.08 × 10−4	3.08 × 10−4	3.07 × 10−4	
	Worst	1.49 × 10−3	3.17 × 10−4	3.08 × 10−4	3.09 × 10−4	
F16	Mean	3.98 × 10−1	4.00 × 10−1	3.98 × 10−1	4.01 × 10−1	
	Std	2.94 × 10−5	2.98 × 10−3	5.74 × 10−4	5.38 × 10−3	
	Best	3.98 × 10−1	3.98 × 10−1	3.98 × 10−1	3.98 × 10−1	
	Worst	3.98 × 10−1	4.08 × 10−1	4.00 × 10−1	4.18 × 10−1	
F17	Mean	3.00	3.00	3.00	3.00	
	Std	8.10 × 10−12	2.08 × 10−8	2.37 × 10−8	2.24 × 10−8	
	Best	3.00	3.00	3.00	3.00	
	Worst	3.00	3.00	3.00	3.00	
F18	Mean	−3.86	−3.85	−3.86	−3.86	
	Std	4.08 × 10−3	2.05 × 10−3	2.93 × 10−3	3.63 × −10−3	
	Best	−3.86	−3.86	−3.86	−3.86	
	Worst	−3.85	−3.85	−3.85	−3.85	

Figure 12 Ablation experiment convergence curve F1–F9.

Figure 13 Ablation experiment convergence curve F10–F18.

The results show that, for most test functions, each method significantly improves the original algorithm, leading to notable enhancements in both convergence speed and optimization accuracy. Furthermore, the combination of these two methods achieves higher efficiency and stronger stability during the optimization process. On the one hand, the improvement in efficiency is reflected in the algorithm’s ability to approach the global optimal solution more quickly, reducing redundant computations and resource consumption. On the other hand, the enhancement in stability ensures consistent and reliable results across multiple runs, even when dealing with complex multimodal problems, effectively avoiding local optima.

Conclusion and outlook

Conclusion

In this study, we propose two innovative optimization strategies: the Entropy Crossover Strategy and the logistic-KNN Inertia Weight. The logistic-KNN inertia weight addresses the limitations of traditional inertia weight frameworks by transitioning from an iteration-based weight assignment to a distance-based framework, where inertia weights are determined based on the distance between individuals and the optimal point. This approach transforms the mapping from “iteration-to-weight” ( t→w) to “distance-to-weight” ( d→w). Additionally, the study introduces “entropy” as an information measure to quantify the information content of parent individuals, refining the traditional random crossover mutation strategy. This enables offspring to inherit advantageous information from the previous generation while retaining a degree of randomness.

To validate the effectiveness of these strategies, they were applied to improve the SHO, resulting in EDSHO. Convergence analysis and benchmarking on 31 test functions selected from CEC2005 and CEC2021 demonstrate that EDSHO outperforms comparison algorithms in terms of convergence speed and solution accuracy, while also exhibiting significant robustness. Statistical analysis using the Wilcoxon rank sum test further confirms the statistical significance of these improvements, and the Friedman rank test quantifies EDSHO’s rankings, consistently placing it among the top-performing algorithms. Moreover, ablation experiments were conducted to assess the contributions of each strategy individually, and the results indicate that both strategies significantly enhance SHO’s performance, further substantiating the effectiveness of the proposed approaches.

Outlook

Given the powerful optimization capabilities of the proposed method, it can be applied not only to more single-objective metaheuristic models but also extended to multi-objective optimization models, with broad applications across various real-world domains. For instance, it can be integrated with deep learning models such as Informer or Transformer to optimize hyperparameters, thereby enhancing performance in time series forecasting, classification, and image recognition. In intelligent scheduling and route planning, this method can optimize logistics delivery routes and vehicle dispatch strategies. In industrial manufacturing, it can refine processing parameters to improve efficiency and reduce costs. In the healthcare field, it can be used for medical image processing and drug molecule optimization. Furthermore, it holds significant potential in energy management, financial model optimization, robotic control, and ecosystem management, offering effective solutions to complex problems and driving technological advancements.

In future research, applying this method to optimize more metaheuristic algorithms and addressing the following critical real-world challenges will be of great importance: (1) optimizing ISCSO engineering structural problems (Kazemzadeh Azad & Kazemzadeh Azad, 2023) to empirically assess the role of the proposed metaheuristic optimization algorithm in structural optimization; (2) optimizing the hyperparameters of neural networks, such as the number of hidden neurons and learning rate, to improve model performance; and (3) extending this method to multi-objective optimization algorithms to further enhance its practical value and application potential. These research directions will be pursued in future work, along with further in-depth discussions.

Supplemental Information

Supplemental Information 1 Algorithm results.

I sincerely thank OpenAI’s ChatGPT-4.0 for its assistance in optimizing the language of my paper.

Additional Information and Declarations

Competing Interests

The authors declare that they have no competing interests.

Author Contributions

Shixing Zheng conceived and designed the experiments, performed the experiments, analyzed the data, performed the computation work, prepared figures and/or tables, authored or reviewed drafts of the article, and approved the final draft.

Data Availability

The following information was supplied regarding data availability:

The code is available at Science Data Bank: Shixing Zheng. 2024. “Code Used by ‘Two Novel Improvement Strategies: Distance-Based Adaptive Inertia Weight and Entropy Crossover Strategy—A Case Study on the Seahorse Optimization Algorithm”. Science Data Bank. doi:10.57760/sciencedb.16238.

The raw data is available in the Supplemental File.

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
