# Peer review of "Improvement strategies for heuristic algorithms based on machine learning and information concepts: a review of the seahorse optimization algorithm"

_PeerJ Computer Science, doi:10.7717/peerj-cs.2805_

## Round 0.1 · original submission · Major Revisions

Dear Authors,

Following a thorough evaluation of the submitted manuscript, it is with regret that we must inform you of its non-acceptance for publication in its present state. A comprehensive review of the pertinent literature is imperative for the manuscript to establish the context and significance of the research. It is recommended that the title of the manuscript include the improvement field. In which field are the "Two novel improvement strategies" proposed? A significant number of the compared methods appear to be metaphor-based, and there is a possibility that they are metaphorical representations of the same method. It is recommended that the experiments be expanded with variants of genetic algorithm and differential evolution.

Best wishes,

Reviewer 1 ·

Basic reporting

The paper proposes two improvement strategies for the Seahorse Optimization Algorithm (SHO). The first strategy introduces a dynamic inertia weight adjustment based on the distance between individuals and the target, allowing a more accurate reflection of relative distances in the optimization process. The second strategy employs an entropy-based crossover mutation technique, enhancing the inheritance of favorable traits from previous generations.

These improvements were tested on CEC2005 benchmark functions, demonstrating significant advancements in convergence speed and solution accuracy compared to traditional methods. The study highlights the effectiveness of these strategies in addressing limitations such as slow convergence and premature stagnation in swarm intelligence algorithms, making a contribution to the optimization research domain.

Experimental design

The study aligns well with the aims and scope of the journal, presenting a research that contributes to the field of optimization algorithms. The research question is clearly defined, relevant, and meaningful, addressing limitations in traditional swarm intelligence algorithms, such as premature convergence and insufficient adaptability. The study effectively identifies a knowledge gap and demonstrates how the proposed strategies—distance-based adaptive inertia weight and entropy-based crossover—contribute to bridging this gap.
Comprehensive descriptions of the methods, including mathematical formulations and algorithmic steps, are provided with sufficient detail to enable replication. The use of benchmark tests and robust statistical analysis further strengthens the reliability and validity of the findings.

Validity of the findings

The article provides an effective solution to key challenges in optimization algorithms and demonstrates strong methodological rigor. The proposed strategies, particularly the distance-based adaptive inertia weight and entropy-based crossover methods, address common issues such as premature convergence and low diversity in existing algorithms.The potential impact and applicability of the proposed strategies in practical, real-world scenarios are can be addresed. Including practical use cases would enhance the relevance of the study.The manuscript does not explore the potential limitations of the proposed strategies or scenarios where their performance might be restricted. Including such a discussion would provide a more balanced perspective.

Reviewer 2 ·

Basic reporting

No comment

Experimental design

No comment

Validity of the findings

No comment

Additional comments

Question 1: What is the approach used to determine the parameter k? Is it selected randomly based on Figure 2, or does it follow a specific rule? Ensuring consistency throughout the paper requires unifying the notation between ski and kwi, avoiding confusion or ambiguity in interpreting the related equations.
Question 2: In equations (21) to (27), the parent weights are described as scalar values. Based on equation (29), this can be understood as a linear combination of the position vectors of the parent individuals. The weights serve to balance the influence between the two parents. According to vector addition rules:
• The offspring's position will be at a distance from the father's position proportional to the magnitude of the mother's vector, scaled by a coefficient.
• Similarly, it will be at a distance from the mother's position proportional to the magnitude of the father's vector, scaled by a coefficient.
In the early stages, when the population has not yet converged, this mechanism may lead to the offspring's position being significantly different from both parents. Does this mechanism contribute meaningfully to exploring the search space and facilitating the optimization process?
Question 3: The results in the paper show that the KNN-Logistics inertial weight strategy and the entropy crossover strategy are highly effective when applied to the SHO algorithm. Accordingly, is there any theoretical or experimental basis to determine whether these strategies would also be effective when applied to other optimization algorithms?
Question 4: When the population converges, the distance between individuals decreases significantly, leading to greater stability in the weight kwi. Is this stability an advantage or a limitation of EDSSO, particularly for problems that require fast convergence? Does this stability affect the algorithm's ability to perform local or global searches in the final stages of the optimization process?
Question 5: In Figure 2, the maximum value wmaxis capped at 0.8 (less than 1). This indicates that the coefficients kwi reduce the influence of the position vectors of individual i, rather than amplifying their influence. Does this reduction limit the individuals' ability to escape local optima? If not, can it be concluded that the effectiveness of EDSSO still largely relies on the original SHO version? Please clarify the contributing factors in this context.

·

Basic reporting

** Place Figures and Tables in the sections in which they are mentioned. **

Remarks on the 1st section:
The citing style throughout the introduction, while consistent, makes the paper difficult to read. I suggest using citing that utilizes parentheses. For example, instead of 'Gu et al. (2022)' write '(Gu et al., 2022)'.
Line 45: after citing Gu et al. it is claimed that Huang (who is not mentioned in (Gu et al., 2022) and can not be found anywhere else in the references) introduced the ASCSO-S algorithm.
Line 47: citation is missing.
Line 49: the algorithm in the reference is called IFEHPSO.
Line 53: the algorithm in the reference is called CGWO.
Line 56: there is no space between the name of the algorithm and the reference. Also, there is no 'Cheng et al.' in the references.
Line 57: there is no space between the name of the algorithm and the reference.
Phrasing issues: the beginning of the sentence in line 45 should be rephrased into 'In (Gu et al., 2022), Gu introduces the ASCSO-S algorithm, that utilizes an S-shaped optimization function [...]'.
Overall introduction remarks: If possible, I would like you to not only write the novel methods introduced but also to tell the readers whether these methods proved to be useful in the studies.
Eq (4): should not be divided into two lines. Also the variable 't' is defined only on line 149 - as with other variables, it should be defined as soon as it is written in an equation or earlier.

Remarks on the 2nd section:
* For multiplication \times and \cdot is used. Only one of them should be used. \times is usually used in simple equations, and \cdot in higher level mathematics. Also, variables that are defined by a single letter don't need multiplication signs, e.g., in eq. (4) (x \times y \times z + ...) should be rewritten to (xyz + ...).
* It is not clearly written that X_{new}^{(1)} adjusts X using movement behavior methodology and X_{new}^{(2)} - using predation behavior methodology.
Line 110: Grammar. [...] generates new individuals that inherit traits [...]

Remarks on the 3rd section:
* Unexplainable gaps between bullet points in between lines 123-128.
Line 143: Rewrite the last sentence so it ends with a dot.
Line 144: In the proposed algorithm, [...].
Line 166: The classic mathematical optimization model [...].
Line 176: '[...] as shown in Equations (21) and (22):'. The same should be done in line 180 after dividing Eq (23) into two. Also Eq (20), Eq (24), Eq (25) needs to be divided into two.
* There is no clear description of Eq (20) given.
* In 3.2 Entropy Corssover Strategy, 4. Normalization of weights: the first sentence and the one on line 187 say the same and are refferencing Eq (27). The current Eq (27) should be written first, then Eq (26) and Eq (25). Descriptions for Eq (26) and Eq (25) are missing.
* Eq (28) the piecewise functions should fit in a single line. If not possible, then when separating it into two or more, the line prior should end in a mathematical operator.
* Line 199: Missing space after the dot.
* Line 200: Bad refferencing to the algorithm pseudocode.
* Figure 1 and 3: Font inconsistency. Also, avoid using the Comic Sans font.

Remarks on the 4th section:
* Line 207: [...] to evaluate the EDSHO algorithm's performance.
* The EDSHO algorithm is renamed to ISHO in Tables 2, 3, 5, 7.
* In lines 230 and 231 the word oscillation might be used inappropriately. Please check its definition and difference compared to the word fluctuate.
* Typo in Figure 7 description.
* Lines 237-239: there is no space between the name of the algorithm and the reference.
* Line 241-242: missing refference to a table, spacing between symbols.
* Line 242: [...] visualization of the algorithms performance.
* Line 250: it is claimed that "NAN" is recorded, but the values in Table 6 say "N/A".
* Figure 8: Names of algorithms should not be cut off.
* Line 256: no need for the dot before '(Significance level: [...]'.

Remarks on the 5th section:
* Line 260: the dot goes after the quatation mark, i.e., [...] Inertia Weight".

Experimental design

In the supplemental pdf file it is stated, that a code 'is now available for reviewers or editors to access', but I could not find any link to the code neither in the peerj website, nor in the manuscript or its references.

Validity of the findings

no comment

Additional comments

Overall, it was interesting reading the paper and the main issues were regarding the formatting of the text and grammar.

Also, the paper analyzes two improvements that are adapted to the SHO algorithm. I wonder whether you have considered only using one of them at a time (either the adaptive inertia weight or entropy crossover) as not always novel approaches keep on improving the algorithms.

---

## Round 0.2 · Minor Revisions

Dear Authors,

Reviewers think that your manuscript needs minor revision. We encourage you to address the minor comments of reviewers and resubmit your paper.

Best wishes,

Reviewer 2 ·

Basic reporting

no comment

Experimental design

no comment

Validity of the findings

no comment

Additional comments

2016–2022, Structures 58, 105409.
ChatGPT đã nói:
The authors have revised the paper; however, some additional comments should be considered:

The CEC2005 benchmark test functions are outdated. The CEC2022 test functions should be considered instead.
The algorithm should be benchmarked on a more challenging standard test suite for truss optimization. To verify the efficiency of the proposed method, all or some of the examples from the following test suite should be tested: "A Standard Benchmarking Suite for Structural Optimization Algorithms: ISCSO 2016–2022, Structures 58, 105409."

·

Basic reporting

Only minor grammar errors, i.e.,
Line 22: missing space at the start of the sentence.
Line 53: add 'the' before ASCSO-S; missing space after comma.
Line 87: missing the end of the sentence.
Line 114: citation separates the word 'performance'.
Line 118: missing space after comma.
Line 157: missing spaces after commas; make t itallic and loose the apostrophes (like in lines 250-251).
Line 259: missing space at the start of the sentence.
Figure 3: Typos: Fathert and Conditiobn
Line 316: missing space at the start of the sentence.
Line 327: missing space at the start of the sentence.
Line 417-418: [...] maintains stable performance [...]
Line 422: [...] conducted the Wilcoxon [...]
Line 448: [...] the Friedman sum rank test [...]

Besides this - everything looks great.

Experimental design

no comment

Validity of the findings

no comment

Additional comments

I am pleased to see that you took the extra time to conduct the ablation experiments, and it was interesting to see how each modification improved the algorithm.

---

## Round 0.3 · accepted · Accept

Dear Authors,

The invited reviewers who requested minor revisions have not responded to the request for your revised manuscript. I have personally evaluated the revised manuscript, and I am satisfied with the current version. The paper is now ready for publication.

Best wishes,